# Effect of Biopesticide Novochizol on Development of Stem Rust *Puccinia graminis* f. sp. *tritici* in Wheat, *T. aestivum* L.

**DOI:** 10.3390/plants13233455

**Published:** 2024-12-09

**Authors:** Andrey B. Shcherban, Ekaterina S. Skolotneva, Anna V. Fedyaeva, Natalya I. Boyko, Vladislav V. Fomenko

**Affiliations:** 1Kurchatov Genomics Center, Institute of Cytology and Genetics SB RAS, Lavrentiev ave. 10, 630090 Novosibirsk, Russia; 2Institute of Cytology and Genetics SB RAS, Lavrentiev ave. 10, 630090 Novosibirsk, Russia; sk-ska@yandex.ru (E.S.S.); fedyaevaav@bionet.nsc.ru (A.V.F.); bojkoni@bionet.nsc.ru (N.I.B.); 3N.N. Vorozhtsov Institute of Organic Chemistry SB RAS, Lavrentiev ave. 9, 630090 Novosibirsk, Russia; chemistnsk@yandex.ru

**Keywords:** wheat, *Triticum aestivum*, stem rust, resistance, biopesticide, chitosan, novochizol, antioxidant activity, reactive oxygen species (ROS), total phenolic content (TPC), peroxidase, catalase

## Abstract

The use of biological plant protection products is promising for agriculture. In particular, chitosan-based biopesticides have become widespread for stimulating growth and protecting plants from a wide range of pathogens. Novochizol is a product obtained by intramolecular cross-linking of linear chitosan molecules and has a globular shape, which provides it with a number of advantages over chitosan. Novochizol has previously been shown to have a stimulating effect on the growth and development of common wheat (*Triticum aestivum* L.). However, the effect of this preparation on the protective mechanisms against rust diseases has not been studied before. Our studies have revealed the dose effect of the preparation on the development of stem rust of wheat. When treating plants with novochizol at a concentration of 0.125% four days before infection, the best results were obtained, namely: a stable reaction was observed and the number of pustules decreased. To identify critical points of the drug’s effect on the protective mechanism against stem rust, we used an adrenaline test, which allows for a quick assessment of the pro/antioxidant status of plant extracts. We also assessed the activity of the major antioxidant enzymes, peroxidase and catalase, using commercial kits and the Folin–Ciocalteu reaction to assess the concentration of phenolic compounds. As a result, two stages were identified in infected plants pretreated with novochizol: early (up to 10 h after inoculation), characterized by antioxidant activity, and late (10–244 h), with prooxidant activity. These stages correspond to two peaks of accumulation of reactive oxygen species (ROS) in response to pathogen infection. The first peak is associated with the accumulation of superoxide anion O_2_−, which is converted into oxygen and hydrogen peroxide under the action of the enzyme SOD (superoxide dismutase). The second peak is associated with the accumulation of H_2_O_2_. Hydrogen peroxide performs a protective function leading to the death of pathogen mycelial cells. In comparison with infected plants without novochizol treatment, we found a decrease in the activity of catalase (an enzyme that breaks down H_2_O_2_) at both stages, as well as peroxidase in the interval from 10 to 144 h after inoculation. Also, an increase in the concentration of phenolic compounds was found in the treated infected plants. We suggest that these changes under the influence of pretreatment with novochizol contribute to enhancements in plant defense functions against stem rust. Taking into account the physicochemical advantages of novochizol over chitosan, which provide a very low effective dose of the drug, the obtained results indicate its promise and safety as a biological plant protection product. This work is a preliminary stage for an extended analysis of the effect of novochizol on plant immunity using biochemical and molecular genetic approaches.

## 1. Introduction

Genetic breeding is a successful for of modern technology for increasing productivity and stress resistance in agricultural plants; however, the development and use of highly effective plant protection products (PPPs) remains relevant. Traditional, chemical pesticides have significant drawbacks: toxicity and allergenicity for humans and animals, reduced nutritional value of food products, development of resistance in harmful organisms, and pollution of the environment. Biological PPPs based on natural compounds are increasingly attracting attention, since, without these drawbacks, they are often not inferior to chemical preparations in effectiveness.

The range of biopesticides and their application methods are quite diverse. But they all target the interaction between the pathogen and the plant, acting either by increasing the plant’s immunity or by interrupting the pathogen’s life cycle. The outcome of this interaction depends on the pathogen, including lifestyle and nutrition strategy: biotrophy, hemibiotrophy, and necrotrophy. In the process of evolution, the host plant has developed hormone-dependent defense mechanisms corresponding to each type of pathogen. Currently, two classical mechanisms are generally recognized: (1) involving salicylic acid (SA) for the host defense response against biotrophic and hemibiotrophic pathogens; (2) involving jasmonic acid (JA) and ethylene, which promote the defense response against necrotrophs and hemibiotrophs [1].

Chitin and chitosan derivatives are widely used as biological PPPs [2,3]. The polymeric carbohydrate chitin is widespread in nature and is a component of the integuments of arthropods (including crustaceans and insects) and fungi. Chitosan is obtained by hydrolysis and deacetylation of chitin. Chitosan-based preparations have a stimulating effect on plant growth and development, and also enhance resistance to abiotic stress [4,5]. Chitosan derivatives are also of particular interest as inducers of resistance to fungal, bacterial, and viral diseases [6,7]. As for biotrophic fungal pathogens, it was shown that chitosan treatment of plants activates the SA-dependent cascade, leading to the accumulation of reactive oxygen species (ROS), hypersensitivity reaction (HSR), synthesis of phenolic substances, and specific protective PR (pathogenesis-related) proteins [8,9]. Increasingly, evidence suggests that other hormones and signaling molecules, especially ROS and nitric oxide (NO), are involved in the regulation of plant defense responses under the action of chitosan [1,8]. The efficiency of chitosan derivatives can be significantly enhanced by their modification: introduction of functional groups of Schiff bases, halogen atoms (Cl or F), metal nanoparticles, urea groups, etc. [7]. A positive protective effect of combining chitosan preparations with other biologically active substances (BASs) and beneficial microorganisms (plant growth-promoting bacteria—PGPB) has been established [10,11,12].

A new chitosan derivative, novochizol, obtained by intramolecular cross-linking of linear chitosan molecules, has great potential. The novochizol molecule has a globular shape, which gives it a number of advantages over chitosan: increased solubility in aqueous solutions, chemical stability and resistance to biodegradation, high penetrating ability and adhesion, as well as the ability to absorb various BASs and slowly release them [13]. The latter property is especially important in terms of creating complex formulations with any BAS, regardless of their structural and physical parameters. Another important difference between novochizol and chitosan is the consistent and identical quality of the product. This allows for both novochizol itself and formulations to be used in a predictable and reproducible manner [14]. Novochizol has a growth-stimulating effect when treated with seeds and leaves. Using common wheat, *T. aestivum*, as an example, it was shown that this preparation enhanced seed germination and contributed to an increase in root mass and the total mass of plants [15]. However, there are still no studies devoted to the influence of novochizol on plant sensitivity to biotic stress factors.

Since chitosan preparations have been previously shown to have a protective effect against various pathogens, it was of interest to evaluate the similar effect of novochizol, as a chitosan derivative. This article is devoted to the laboratory evaluation of the effect of novochizol on the susceptibility of common wheat plants to stem rust caused by *P. graminis* f. sp. *tritici* (Pgt). The latter is a very harmful fungal pathogen that in some years affects up to 40% of wheat crops in Russia [16]. Since *P. graminis* is a biotrophic pathogen, we hypothesize that novochizol enhances wheat resistance to stem rust through an SA-dependent cascade that affects the antioxidant system (see above). This system is known to maintain the level of ROS required for protection against the pathogen and serves as a convenient marker to assess the action of biopesticides [17]. At the initial step, it was necessary to find the critical points at which the reaction to the pathogen occurs under the influence of novochizol. For this purpose, biochemical tests were used in this work: an adrenaline test for antioxidant activity and an assessment of peroxidase and catalase activity. Additionally, an express test for phenolic compounds was used. In the future, a more detailed study of the effect of novochizol on wheat immunity will be carried out with an analysis of the expression of various known PR genes and cytophysiological studies.

## 2. Results

### 2.1. Visual Assessment of Novochizol’s Effect on Stem Rust Development

During the first stage, the effect of different novochizol concentrations on the disease development was studied on the susceptible variety Novosibirskaya 29. A wide range of drug concentrations was used in the experiments—from 0.125 to 2.5%. Pustules with IT “4” (pustules of type 4) were formed on the plants treated with water (k). The results showed that novochizol in any concentration affected the development of the disease, which was manifested in a decrease in the density of urediniopustules, a reduction in their size, and the appearance of chlorosis zones of various sizes around sporulation (Figure 1). The minimal IT decrease was noted when using 2.5% solution of novochizol (IT “3”,”3+”). The preparation in concentrations of 0.125, 0.75, and 1.5% induced a stable reaction of plants. The maximum number of pustules was observed on the leaves of control plants, but the minimum was found when treating plants with novochizol at a dose of 0.125% (20 and 8 pustules per leaf, respectively) (Figure 1). This concentration was used for further study of plant defense reactions.

### 2.2. AA Determination

As described in Materials and Methods, AA was assessed by the degree of inhibition of superoxide radical by plant extracts in the adrenaline autooxidation reaction. The AA level of control plants changed from prooxidant activity at 2–10 hp/in to antioxidant activity (72–240 hp/in) (Figure 2A). We assume that this change was caused by stress due to the limited size of the pot and the root ball. Uninfected novochizol(+) plants showed increased antioxidant activity up to 24 hp/in compared to control, which may be due to the effect of novochizol (Figure 2A; Appendix A). In infected plants, compared to the control, we observed an increased level of AA at 2–10 hp/in, most pronounced in plants not treated with novochizol at 7 hp/in. After 10 hp/in, the curve shifts towards prooxidant activity, especially in plants pretreated with novochizol (Figure 2A).

### 2.3. TPC Determination

TPC was assessed using Folin–Ciocalteu reagent (see Section 4). On average, control plants have a higher TPC value compared to other groups, except for the 24 hp/in point (Figure 2B) (Figure 2B; Appendix A). Infected novochizol(−) plants demonstrated a lower TPC value at the interval about 5–144 hp/in, compared to infected novochizol(+) plants.

### 2.4. Estimation of Peroxidase and Catalase Activity

The level of catalase (CAT) activity in uninfected control and novochizol(+) plants remained more or less stable during the experiment (Figure 2C; Appendix A). In infected novochizol(−) plants, catalase activity increased at 10 hp/in and then decreased to control level at 72 hp/in. After the last point, the activity increased again, reaching its highest value at 144 hp/in. Unlike the previous group, no significant changes in the level of catalase activity were observed in the infected novochizol(+) plants compared to control plants, with the exception of a slight decrease in this level below the control at 72 hp/in.

In uninfected novochizol(+) plants at 24 hp/in, a significant increase in peroxidase (POD) activity was observed relative to the control (Figure 2D; Appendix A). But after 2 days, the activity dropped below the control level. In infected plants, a reduced activity of this enzyme was observed within about 2–10 hp/in. After that, the infected novochizol(−) plants demonstrated an increase in peroxidase activity up to 144 hp/in, whereas novochizol(+) plants showed a later increase in this activity after 72 hp/in reaching the level of the previous group at 144 hp/in.

## 3. Discussion

This paper presents preliminary data on the effect of novochizol on the development of stem rust in common wheat. A concentration-dependent effect of novochizol on the development of stem rust was established (Figure 1). This is consistent with the results of previous studies on the effect of chitosan doses on defense responses [4]. In the variant with 0.125% novochizol treatment, the best ratio of pustules by infectious type and condition of leaves was revealed. We used this preparation in the following stages of work.

In the 2000s, a hypothesis of a two-level organization of plant immunity was formulated, which was called “PTI-ETI” [18]. It was assumed that plants have PRRs (pattern recognition receptors) that recognize molecules of pathogenic microorganisms (PAMPs—pathogen-associated molecular patterns). As a result of recognition of these molecules, the first level of PTI (PAMP-triggered immunity) protection is launched. After overcoming PTI, the second level of protection associated with the recognition of specific effectors is activated—ETI (effector-triggered immunity).

According to earlier studies, two peaks of ROS formation in response to pathogen infection were revealed in plants. The first peak occurs within minutes or few hours after recognition of PAMPs and was associated with activation of the enzyme NADPH oxidase, which constitutively exists in the membrane, and formation of superoxide anion O_2_−, which is quickly converted into H_2_O_2_ by the enzyme superoxide dismutase (SOD) [19]. To study the pro/antioxidant activity associated with SOD, we used the adrenaline test (www.csun.edu/~hcchm001/sodassay.pdf accessed on 5 December 2024) [20,21]. As a result, we observed an increase in AA at 2–10 hp/in, which corresponds to the first peak of ROS accumulation (Figure 2A). A higher level of AA is characteristic of infected plants without treatment with novochizol, while the level of infected novochizol(+) plants is almost the same as that of uninfected novochizol(+) plants. This may indicate a lower level of stress sensitivity provided by the novochizol treatment. After 10 hp/in, we observed a decrease in AA in plants, both treated and untreated with novochizol (prooxidant activity).

The second peak appears after 3–5 days and is associated with de novo synthesis of enzymes of the pro/antioxidant system (peroxidases, catalase, oxalate oxidases, etc.). The pro/antioxidant system maintains an optimal level of ROS in tissues. Catalase breaks down H_2_O_2_ into water and molecular oxygen; various peroxidases, polyphenol oxidase, and ascorbate oxidase utilize ROS in oxidative reactions [22]. Next, we analyzed the activity of antioxidant enzymes peroxidase and catalase.

The level of catalase activity in infected plants without novochizol treatment demonstrated two peaks at 10 and 144 hp/in (Figure 2C). These peaks probably correspond to the main peaks of ROS accumulation. In infected plants with treatment, this level was generally lower, except for the point 24 hp/in. Along with the data on the AA estimation (see above), this may indicate a higher stress resistance of the latter group. As for peroxidase, its activity level was lower in the same group in the period from 10 to 144 hp/in (Figure 2D). During the assessment of the effect of complex chitosan preparations on wheat resistance to fungal diseases, the ability of these preparations to inhibit the activity of catalase and peroxidase was shown [23]. Suppression of the activity of these enzymes leads to an increased level of H_2_O_2_ and stimulation of defense reactions against pathogens. It is known that ROS, including hydrogen peroxide, are directly involved in the formation and implementation of plant defense responses when infected with pathogens. The accumulation of H_2_O_2_ was manifested in the suppression of the development of the biotroph *Puccinia recondita* and the suppression of disease symptoms in the form of necrosis for the hemibiotroph *Cochliobolus sativus* [23]. We hypothesize that a similar mechanism operates against stem rust in wheat tissues treated with novochizol. Recently, a histochemical analysis of this plant material was carried out at 96 hp/in and an intensive accumulation of H_2_O_2_ in the leaves was established [24]. Apparently, this is associated with the manifestation of the second peak of the oxidative burst. Moreover, in areas of intensive accumulation of H_2_O_2_, the death of pathogen mycelial cells was observed. A protective role of H_2_O_2_ against fungal pathogens has also been confirmed by other authors [25].

Phenolic substances play an important role in protecting plants from pathogens [26]. It was previously shown that the formation of phenols and strengthening of cell walls with lignin after treatment with chitosan were the most typical defense reactions against pathogenic fungi [4,6]. Our data show that treatment with novochizol stimulated more intensive accumulation of phenols than in untreated plants throughout most of the experiment (Figure 2B). This difference is most noticeable at an early stage up to 10 hp/in.

It was previously shown that after the use of chitosan in plants infected with fungi, the expression of genes encoding PR proteins (chitinase, glucanase, peroxidase, polyphenoloxidase, PR-1, PR-5, etc.) increased [8,9,27,28]. Some of these PR proteins, like peroxidase and polyphenoloxidase, are involved in the metabolism of phenolic compounds. We further plan to study the expression of various PR genes in infected wheat treated with novochizol.

It seems interesting to compare the efficiency of novochizol with other chitosan- based preparations in field conditions. In the Russian Federation, a number of these preparations containing PGPB (Vitaplan, Gamair), as well as complex preparations with organic acids chitosan I and II, have been developed [29]. Comparison of their effects with the effect of novochizol on the development of root rot in wheat showed similar efficiency (unpublished data). Unfortunately, we were unable to properly evaluate the effect of novochizol on stem rust in the field due to unfavorable conditions for this disease in the last 2 years. Nevertheless, novochizol demonstrated a more pronounced effect on grain productivity, compared with the above chitosan preparations [29,30].

Thus, we have identified for, the first time, the critical points associated with the effect of novochizol in response to infection of wheat with the fungal pathogen of stem rust. The identified changes in pro/antioxidant activity, activity of the main antioxidant enzymes, and concentration of phenolic compounds indicate the similarity of the protective effects of novochizol and some other forms of chitosan. However, given the cost-effectiveness provided by the low active concentration and ease of application of novochizol, one can assume its high potential as a safe and effective biological plant protection product.

## 4. Material and Methods

### 4.1. Plant Material

The research objects were 10-day-old seedlings of the common wheat variety Novosibirskaya 29, susceptible to stem rust. The plants were grown in vessels with soil, which is recommended for experiments with rust fungi by international protocols [31].

### 4.2. Stem Rust Pathogen

To infect the seedlings, we used urediniospores of the West Siberian population *Pgt* from the collection of the Institute of Cytology and Genetics of SB RAS. Urediniospores were stored at −70 °C before the experiment and re-cultivated on the susceptible variety of common wheat “Khakasskaya” [32].

### 4.3. Novochizol Treatment and Plant Infection

Novochizol™ base, having a degree of deacetylation of 90% or above and a molecular weight in the 500 kD range, was provided by NOVOCHIZOL SA, Monthey, Switzerland (www.novochizol.ch accessed on 5 December 2024). Novochizol aqueous suspensions were obtained as described in [15]. In laboratory conditions, greater biomass accumulation and more uniform seedling development were observed with 0.1% novochizol treatment [15]. However, the effect of any concentration of novochizol on the development of stem rust has not been previously assessed, so we selected the range 0.1–2.5%, since concentrations above 2.5% complicate both the processes of dissolution and application of the preparation by spraying. The seedlings were treated with novochizol solutions at concentrations of 0.125, 0.75, 1.5, and 2.5%. The solutions were applied to the plants using a sprayer (15 mL/100 plants) four days before infection with *Pgt*.

A suspension of urediniospores with a concentration of 0.8 mg/mL Novec 7100 (3MNovec^TM^, St. Paul, MN, USA) [33] was applied to the seedlings using a sprayer. Control plants were sprayed with bidistilled water. Inoculated plants were incubated for 24 h in a humidified chamber in the dark at a temperature of 15–20 °C to ensure maximum spore germination. Then, the plants were transferred to growth chambers and incubated under 16 h illumination with phytolamps with an intensity of 10,000 lux at a temperature of 26–28 °C. This temperature is necessary for the formation of appressoria, penetration of the pathogen into the stomata, and the development of infectious hyphae in the intercellular spaces of the plant [34].

### 4.4. Evaluation of the Susceptibility Level of Plants to Infection

The disease scoring was assessed in accordance with the recommendations of Roelfs et al. (1992) for seedlings [34]. Fourteen days after inoculation, we evaluated both the infection type and the number of pustules on the leaf blade. The infection types of plant response (IT) was determined 12–14 days after inoculation using the modified Stackman scale, where IT “0”, “;”, “1”, and “2” were interpreted as resistant (low—L), and “3”, “3+”, and “4” were interpreted as susceptible (high—H) [35].

### 4.5. Evaluation of Antioxidant Activity of Plant Extracts

Taking into account the previous assessment of the effect of novochizol on disease development, biochemical analyses were performed on the plants treated with 0.125% novochizol. The material was collected 2, 7, 10, 24, 72, 144, and 240 h after inoculation (hp/in). Leaf pieces were frozen in liquid nitrogen and stored at −80 °C until analysis. Extracts were obtained using a modified method [36]. Frozen leaves were ground in cooled sterile mortars. A quantity of 0.1 g of tissue was placed in cooled Eppendorf tubes and 0.6 mL of cooled (4 °C) 0.2 M K-phosphate buffer pH 7.8 was added. The tubes were thoroughly shaken and centrifuged for 20 min at 10,000 rpm at 4 °C. The supernatant was collected and stored on ice until analysis. Antioxidant activity (AA) of plant extracts was assessed by their ability to inhibit the reaction of adrenaline autooxidation with its conversion to adrenochrome, according to a modified method [24]. The reaction mixture was obtained by sequentially adding 0.045 mL of 0.2 M K-phosphate buffer (pH 7.8) and 0.15 mL of 0.1% (5.46 mM) adrenaline hydrochloride solution to 3 mL of 0.2 M Na-carbonate buffer (pH 10.65). The control sample (D1) was mixed, placed in a spectrophotometer SPEKS SSP 705 (“Spectroscopic Systems”, Moscow, Russia), and the optical density was measured for 30 s. at a wavelength of 347 nm (absorption spectrum of adrenochrome). When analyzing plant extract samples, the K-phosphate buffer was replaced with 0.045 mL of the extract (D2). AA (%) was assessed as a percentage of adrenaline autooxidation inhibition using the following formula: AA = ((D1 − D2) × 100)/D1.

### 4.6. Determination of Total Phenolic Content

Total phenolic content (TPC) was estimated using Folin–Ciocalteu reagent [37]. Frozen leaves were ground in cooled sterile mortars. Samples of 0.02 g of ground tissue were placed in cooled Eppendorf tubes; 2 mL of ice-cold 95% methanol was added in each sample, vortexed, and left in the dark for 48 h at room temperature for extraction. The tubes were centrifuged at 14,500 rpm for 5 min and the supernatant was transferred to new tubes. A quantity of 0.1 mL of each sample was used for the analysis. Standard solutions of gallic acid and 95% methanol for the blank probe were used to construct the calibration curve. Gallic acid solutions were prepared in 95% methanol; the final concentrations were 10, 50, 100, 150, 200, and 240 μg/mL; 0.2 mL of 10% (*v*/*v*) Folin–Ciocalteu reagent was added to the analyzed samples and mixed in a vortex, and after 3–4 min, 0.8 mL of 700 mM Na_2_CO_3_ was added. Immediately after that, the samples were vortexed and left in the dark at room temperature for 2 h. After 2 h, the samples were centrifuged as described above. The optical density of the solutions was determined at λ = 765 nm using a spectrophotometer. TPC was determined from a calibration graph using the regression equation between gallic acid standards and optical density of solutions at λ = 765 nm and expressed as gallic acid equivalents.

### 4.7. Evaluation of Enzymatic Activity

The collected plant material was stored until analysis at −80 °C as described above. Samples were ground in cooled mortars, 0.05 g was collected, and 0.45 mL of cold PBS buffer (pH 7.4) with 1 mM EDTA was added. The samples were then centrifuged at 7000 rpm for 10 min at 4 °C. The supernatant was collected and stored on ice until analysis. The analyses were performed on the same day. The protein level in the samples was estimated using the Bradford protein determination kit (Catalog No: E-BC-K168-S, Elabscience Inc., Houston, TX, USA). No sample dilution was required to determine catalase activity. To determine peroxidase activity, the samples were additionally diluted with 0.01 M PBS buffer (pH 7.4) containing 1 mM EDTA with a dilution factor of 1:10. The enzymatic activity of each sample was determined according to the manufacturer’s protocols using the Plant Peroxidase (POD) Activity Assay Kit (Catalog No.: E-BC-K227-S) and Catalase (CAT) Activity Assay Kit (Catalog No.: E-BC-K031-M) from Elabscience Inc.

### 4.8. Statistical Analysis

All experiments were conducted in 3 biological and 2 analytical replicates. Data obtained from each analysis were first evaluated for normality. Data were analyzed using Statistica 12 software from StatSoft (Tulsa, OK, USA). To test for the significant difference between the two samples, we used the Mann–Whitney U test (Appendix A). Figure 2 presents the mean values and their standard errors.

## Figures and Tables

**Figure 1 plants-13-03455-f001:**
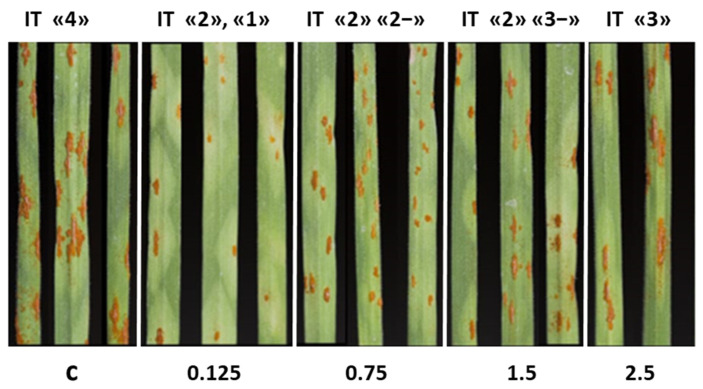
Visual assessment of the novochizol effect on the stem rust development. c—control plants treated with water; 0.125; 0.75; 1.5; 2.5—the different concentrations of novochizol (in %) used to treat plants before infection with stem rust pathogen. The sensitivity indices of IT are indicated above the figure.

**Figure 2 plants-13-03455-f002:**
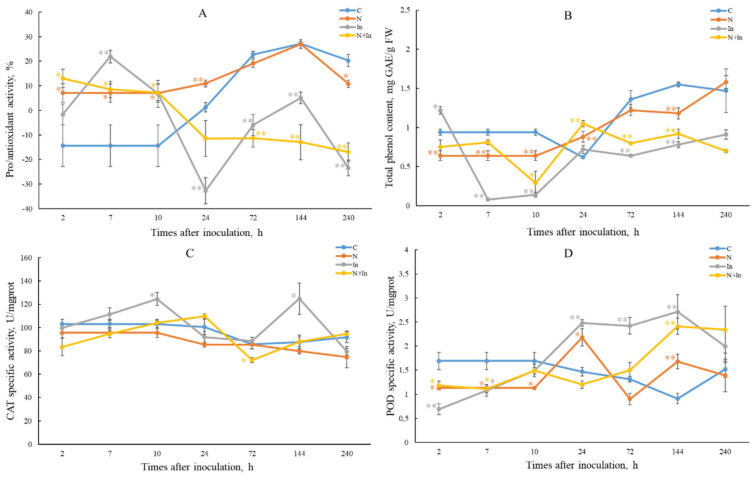
Changes in pro/antioxidant activity (**A**), total phenol content (**B**), catalase activity (**C**) and peroxidase activity (**D**) in leaves of common wheat after treatment with 0.125% novochizol and inoculation with urediniospores of the West Siberian population *Pgt*. Designations: C—control; N—novochizol treatment; In—inoculation; N+In—novochizol treatment and inoculation. *p* values * *p* ≤ 0.05, ** *p* ≤ 0.01, when compared with control plants at the same time point.

## Data Availability

The data presented in this study are available upon request from the corresponding authors.

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
