# Peer review of "Effect of Biopesticide Novochizol on Development of Stem Rust Puccinia graminis f. sp. tritici in Wheat, T. aestivum L."

_plants, 2024, doi:10.3390/plants13233455_

Round 1
Reviewer 1 Report
Comments and Suggestions for Authors
This ms. Reported a chitosan derivatives on wheat stem rust. I have the following suggestions:
1. Provide more information and/or reference about novochizol . Line 77-81 donot has real data about it.
2. Many chitosan derivatives are previously studied for control of different plant diseases two decades ago by effect on plant host or induction of resistance. What is the novelty or difference fornovochizol ?
3. 3.1. Visual assessment of the novochizol effect on the stem rust development and Figure 1. , data for stem rust such as disease index should be provided.
4. Fig.1. 0,125; 0,75; 1,5; 2,5 – the different concentrations of novoch. Not learR
5. Figure 2. Difference between treatments should be marked.
6. Results of anti- oxidant activity, reactive oxygen species (ROS),total phenolic content (TPC),peroxidase,catalase at different time point. Their difference?association and relationship with novochizolshould be presented more clear and professional
7. Discussion should focus on the main novelty of this study. Please do not say much about common sense about chitosan derivatives in plant disease control.
Author Response
First of all, many thanks for such a thorough analysis of our article. We tried to take into account all the comments during the revision process, which will undoubtedly make the article better.
- Provide more information and/or reference about novochizol . Line 77-81 donot has real data about it.
- Many chitosan derivatives are previously studied for control of different plant diseases two decades ago by effect on plant host or induction of resistance. What is the novelty or difference for novochizol?
First, we have added сharacteristics of novochizol preparation in Materials and Methods (lines 134-137). Also, we have added some references [13,14] that can confirm the most important features of novochizol (Introduction, lines 91-100). It should be noted that so far a few works on Novochizol has been published in agrobiology, and in the area of plant protection our work is pioneering.
- 3.1. Visual assessment of the novochizol effect on the stem rust development and Figure 1. , data for stem rust such as disease index should be provided.
Since the routine disease index is useless for seedling studies in laboratory, we made disease scoring as it was recommended in Roelfs et al., 1992 (Roelfs, A.P.; Singh, R.P.; Saari, E.E. Rust diseases of wheat: concepts and methods of disease management. Cimmyt, Mexico DF, Mexico. 1992, 81). (See Materials and Methods, Reference 17).
We estimated both infection type and number of pustules per leaf blade. We added these data into updated paper (lines in Material and Methods; lines in Results) and put the IT indices for each concentration in Figure 1.
- Fig.1. 0,125; 0,75; 1,5; 2,5 – the different concentrations of novoch. Not learR ??
- Figure 2. Difference between treatments should be marked.
Taking into account your remark, we have highlighted with asterisks in the figure the points where the most reliable differences were observed. * p ≤ 0.05, ** p ≤ 0.01, when compared with control plants at the same point. In this case, Table 1 can be omitted (or transferred to Supplementary Materials).
- Results of anti- oxidant activity, reactive oxygen species (ROS),total phenolic content (TPC),peroxidase,catalase at different time point. Their difference?association and relationship with novochizolshould be presented more clear and professional.
Sorry, what form of presentation do you consider as the most professional? We thought that the presentation of data in dynamics is the most illustrative form.
- Discussion should focus on the main novelty of this study. Please do not say much about common sense about chitosan derivatives in plant disease control.
Thanks for remark! Since novochizol is a modified form of chitosan, it would be justified to compare our data with similar results on the effects of other forms of chitosan. Moreover, there are no other comparative studies yet that would study the effect of novochizol at the biochemical level on any fungal plant pathogens. However, in the introduction we have expanded the information about novochizol to emphasize the prospects of its use and have rewritten the last part of discussion to emphasize the novelty and significance of our research (lines 349-365).
Reviewer 2 Report
Comments and Suggestions for Authors
The solutions should be present in umol ionstead of percentage. Figure is too small in the same page, if the Authors separate them it will more clear for the readers.
Reviewer 3 Report
Comments and Suggestions for Authors
The use of Novochizol, a novel chitosan derivative, is a significant advancement in biological plant protection strategies. However, the study's novelty and broader implications could be more explicitly articulated in the introduction and discussion. Specifically:
Expand on how Novochizol compares to other biopesticides in terms of efficacy, cost, and environmental impact.
Highlight why this research is critical given the current challenges in sustainable agriculture (e.g., growing resistance to chemical fungicides, climate change).
The study's objectives are somewhat fragmented across sections. In the introduction, clearly articulate:
The primary and secondary research goals.
Hypotheses being tested (e.g., does Novochizol enhance wheat resistance to stem rust, and by what mechanisms?).
Improve the logical flow:
Merge overlapping sections in the discussion to streamline key findings.
Ensure all conclusions are directly supported by results.
While the abstract is informative, it lacks a clear summary of the broader implications of the findings. Conclude with how these findings could influence future plant protection strategies or contribute to sustainable agriculture. Mention the specific methodologies used (e.g., adrenaline test, enzyme assays) briefly to enhance scientific rigor.
The introduction provides a solid foundation, however, the justification for studying Novochizol could be more detailed:
Discuss the limitations of current biopesticides and how Novochizol addresses these gaps.
Elaborate on why stem rust is a particularly compelling target.
Include a schematic or conceptual framework outlining the hypothesized interaction between Novochizol treatment, ROS dynamics, and phenolic compound accumulation.
Clarify why specific concentrations of Novochizol were chosen, particularly the wide range (0.125% to 2.5%).
Discuss observed phenomena (e.g., changes in catalase activity) in the context of plant-pathogen interaction theories.
Explore deeper into the molecular mechanisms underlying Novochizol's effects: How does it enhance phenolic compound production? What specific genes or pathways might be involved?
While the discussion references related research, provide a comparative table summarizing: Findings of previous studies on chitosan derivatives; Novochizol's relative efficacy in enhancing ROS accumulation and defense enzyme activity,
Emphasize how these results could inform: Field-level applications (e.g., optimal application methods and timing for Novochizol); Integration into existing integrated pest management (IPM) frameworks.
Conclude with a stronger emphasis on the environmental and economic advantages of Novochizol.
Provide a clear recommendation for scaling this research to practical agricultural settings.
Author Response
First of all, many thanks for such a thorough analysis of our article. We tried to take into account all the comments during the revision process, which will undoubtedly make the article better.
The use of Novochizol, a novel chitosan derivative, is a significant advancement in biological plant protection strategies. However, the study's novelty and broader implications could be more explicitly articulated in the introduction and discussion. Specifically:
Expand on how Novochizol compares to other biopesticides in terms of efficacy, cost, and environmental impact.
Thanks for remark! In the introduction we have expanded the information about the effectiveness of novochizol (lines 91-104) to emphasize the prospects of its use and have rewritten the final part of discussion (lines 349-365) to emphasize the novelty and significance of our research. As for cost efficiency, it is not yet possible to assess it until all tests of the drug have been carried out and technologies for obtaining and application of novochizol for widespread use in agriculture have been developed.
Highlight why this research is critical given the current challenges in sustainable agriculture (e.g., growing resistance to chemical fungicides, climate change).
At the beginning of the Introduction we mentioned why the development of biological control agents is important and what advantages they provide compared to chemical pesticides (lines 55-62). With regard to novochizol specifically, we have expanded the information about this drug to make it clearer why its study is important. It should be noted that so far a few works on Novochizol has been published in agrobiology, and in the area of plant protection our work is pioneering.
The study's objectives are somewhat fragmented across sections. In the introduction, clearly articulate:
The primary and secondary research goals.
We have changed the Introduction to make the initial and final goals of the study clearer (lines 105-121).
Hypotheses being tested (e.g., does Novochizol enhance wheat resistance to stem rust, and by what mechanisms?).
We have presented our hypothesis (lines 110-114).
Improve the logical flow:
We have reformatted some paragraphs in Introduction to keep logical flow
Merge overlapping sections in the discussion to streamline key findings.
We have reformatted all the discussion
Ensure all conclusions are directly supported by results.
We hope so
While the abstract is informative, it lacks a clear summary of the broader implications of the findings. Conclude with how these findings could influence future plant protection strategies or contribute to sustainable agriculture. Mention the specific methodologies used (e.g., adrenaline test, enzyme assays) briefly to enhance scientific rigor.
We have added a conclusion to the abstract: “Taking into account the physicochemical advantages of novochizol over chitosan, which provide a very low effective dose of the drug, the obtained results indicate its promise and safety as a biological plant protection product.” And mentioned the methods used (lines 26-30).
The introduction provides a solid foundation, however, the justification for studying Novochizol could be more detailed:
Discuss the limitations of current biopesticides and how Novochizol addresses these gaps.
We gave the information about advantages of novochizol (lines 91-100).
Elaborate on why stem rust is a particularly compelling target.
We have added this (lines 109-110).
Include a schematic or conceptual framework outlining the hypothesized interaction between Novochizol treatment, ROS dynamics, and phenolic compound accumulation.
This article is presented as a brief report. Taking into account your comments, we have to additionally increase its volume. Besides, it is not entirely clear how to display completely different variable parameters on one scheme. If you know such a way, please, let us know. We will do it.
Clarify why specific concentrations of Novochizol were chosen, particularly the wide range (0.125% to 2.5%).
In laboratory conditions, greater biomass accumulation and more uniform seedling development were observed with 0.1% novochizol treatment [ ]. However, the effect of any concentration of novochizol on the development of stem rust has not been previously assessed, so we selected the range 0.1-2.5% since concentrations above 2.5% complicate the processes of dissolution and application of the preparation by spraying. Lines 137-142
Discuss observed phenomena (e.g., changes in catalase activity) in the context of plant-pathogen interaction theories.
We discuss this (lines 321-336).
Explore deeper into the molecular mechanisms underlying Novochizol's effects: How does it enhance phenolic compound production? What specific genes or pathways might be involved?
We have added this information (lines 337-348).
While the discussion references related research, provide a comparative table summarizing: Findings of previous studies on chitosan derivatives; Novochizol's relative efficacy in enhancing ROS accumulation and defense enzyme activity,
Ok, we give an example such a table in Suppl. But we should ask the editor if it is possible to expand the article above limits, since this table contains a number of references. Besides, it is hard to compare the data from different studies accurately, since different materials and methods are used. Works with novochizol of this type, except ours, has not yet been carried out.
Emphasize how these results could inform: Field-level applications (e.g., optimal application methods and timing for Novochizol); Integration into existing integrated pest management (IPM) frameworks.
We think that it is too early to say about field level application and integration our results into practice
Conclude with a stronger emphasis on the environmental and economic advantages of Novochizol.
We have changed the end of discussion to emphasize the environmental safety and economic advantage of Novochizol.
Provide a clear recommendation for scaling this research to practical agricultural settings.
Since it takes more than one year to test a drug in field conditions, we cannot provide such recommendations yet.
Round 2
Reviewer 1 Report
Comments and Suggestions for Authors
Most of my comments are responsed in this revised version. Therefore, I suggest accept it.
Reviewer 3 Report
Comments and Suggestions for Authors
ok